# A Compact Modified Two-Arm Rectangular Spiral Implantable Antenna Design for ISM Band Biosensing Applications

**DOI:** 10.3390/s23104883

**Published:** 2023-05-18

**Authors:** Mustafa Hikmet Bilgehan Ucar, Erdem Uras

**Affiliations:** 1Information Systems Engineering Department, Kocaeli University, Kocaeli 41001, Turkey; 2Turkish Airlines Technic Inc., Turkish Airlines, Istanbul 34912, Turkey

**Keywords:** biosensing, implantable antenna, ISM band, microstrip antenna, rat skin, rectangular spiral

## Abstract

This paper presents a new microstrip implantable antenna (MIA) design based on the two-arm rectangular spiral (TARS) element for ISM band (Industrial, Scientific, and Medical 2.4–2.48 GHz) biotelemetric sensing applications. In the antenna design, the radiating element consists of a two-arm rectangular spiral placed on a ground-supported dielectric layer with a permittivity of $\epsilon_{r}$ = 10.2 and a metallic line surrounding this spiral. Considering the practical implementation, in the proposed TARS-MIA, a superstrate of the same material is used to prevent contact between the tissue and the metallic radiator element. The TARS-MIA has a compact size of 10 × 10 × 2.56 mm^3^ and is excited by a 50 Ω coaxial feed line. The impedance bandwidth of the TARS-MIA is from 2.39 to 2.51 GHz considering a 50 Ω system, and has a directional radiation pattern with directivity of 3.18 dBi. Numerical analysis of the proposed microstrip antenna design is carried out in an environment with dielectric properties of rat skin (Cole–Cole model $\epsilon_{f}$ (ω), ρ = 1050 kg/m^3^) via CST Microwave Studio. The proposed TARS-MIA is fabricated using Rogers 3210 laminate with dielectric permittivity of $\epsilon_{r}$ = 10.2. The in vitro input reflection coefficient measurements are realized in a rat skin-mimicking liquid reported in the literature. It is observed that the in vitro measurement and simulation results are compatible, except for some inconsistencies due to manufacturing and material tolerances. The novelty of this paper is that the proposed antenna has a unique two-armed square spiral geometry along with a compact size. Moreover, an important contribution of the paper is the consideration of the radiation performance of the proposed antenna design in a realistic homogeneous 3D rat model. Ultimately, the proposed TARS-MIA may be a good alternative for ISM-band biosensing operations with its miniature size and acceptable radiation performance compared to its counterparts.

## 1. Introduction

Today, with the use of various devices such as smart watches, health bands, and fitness tracking by millions of users worldwide, the place and importance of wearable technologies in our lives are increasing daily. However, in the near future, it is predicted that wearable devices will be replaced by new generation smart implantable systems placed under the skin, where health and personal data monitoring experiences are further customized [1]. As an example of the existing implantable systems used in biomedical applications today, Bluetooth-controlled hearing aids, pacemakers, insulin pumps, RFID tags placed under the skin, and deep brain stimulation can be listed [2,3,4]. With various mobile sensors in implantable devices, individuals’ movements and sleep can be monitored while they maintain their daily routines, and physiological parameters can be recorded throughout the day. In this way, implant technologies are becoming an indispensable part of human life for various purposes, from providing a healthy life to diagnosing and treating diseases [5,6,7].

Physiological parameters are performed by biosensing, whose main purpose is to detect and quantify the presence of certain target molecules or analytes in a sample [8]. Biosensors can be designed to detect a wide range of targets, including proteins, nucleic acids, carbohydrates, chemicals, and even whole cells or microorganisms. By providing a rapid, sensitive, and selective means of detecting target molecules, biosensors have many practical applications in healthcare, food safety, environmental monitoring, and biodefense [8]. In healthcare, biosensors can be used for early detection and diagnosis of diseases, monitoring of therapeutic drug levels, and detection of infectious agents. Biosensing systems can be quite complex and require expertise in multiple fields, including biology, chemistry, engineering, and computer science. However, these systems have tremendous potential for various applications, including medical diagnostics, environmental monitoring, and food safety testing. Biosensing systems typically have several key components: biological sensing elements, transducers, signal amplification and processing components, power supply, and communication units [8,9,10].

However, before being used by humans, the developed systems must be tested using laboratory animals (such as rats), reconfigured if necessary, and/or optimized to eliminate potential side effects [11]. The design of implantable systems is limited by several specific system requirements, such as biocompatibility and miniature size [12,13,14,15,16,17,18,19,20,21]. The development of biomedical implantable devices is directly related to developments in IC, robotics, power supplies, and RF technologies [5,6,7]. Antenna structures, the most critical component of the communication unit, are an important aspect of biosensing systems because they can significantly impact the system’s performance [8,9,10]. In particular, antenna design plays a critical role in the ability of the system to detect and measure the signal from the biological sensing element. Antennas are responsible for receiving and transmitting the signal; therefore, their design is crucial to ensure that the signal can be detected and measured accurately. The sensitivity of a biosensing system is directly related to the ability of the antenna to capture and transmit the signal. A well-designed antenna can improve the sensitivity and accuracy of the biosensing system. The antenna design can help minimize the noise level and improve the signal-to-noise ratio, thereby enhancing measurement accuracy [8,9,10]. Biosensing systems typically require integration with other components, such as transducers and amplifiers. Antenna design can facilitate the integration process and improve the system’s performance. Overall, antenna design is an important aspect of biosensing systems. Care must be taken to ensure that the size is optimized for the application so that the design meets the system requirements while realizing intra-body implantation. In particular, the production of small-sized implant devices depends on the miniature design of the antenna elements that provide the wireless communication of the system. Thus, researchers are carrying out studies to reduce the antenna size by taking advantage of the innovations brought by the technology [11,13,14,15,16,17,18,19,20,21,22,23,24,25,26,27,28,29,30,31,32,33,34,35,36,37,38].

In the literature, the realization of measurements by placing an implantable antenna in a rat skin-mimicking liquid was first revealed in 2009 by the study of T. Karacolak et al. In the study, a microstrip implant antenna design operating in MICS (Medical Implant Communication Services 402–405 MHz) and ISM (Industrial, Scientific, and Medical 2.4–2.48 GHz) bands with dimensions of 23 × 23 × 5 mm^3^ have been proposed. In addition, meander lines were used to miniaturize the antenna, and a short circuit pin was added [27]. In the study conducted by A. Kiourti et al., two different patch antennas with circular geometry designed according to the electrical properties of rat skin that can operate in the MICS band were presented in 2013. The first of these antennas is a double-layered cylindrical structure with a 12 mm diameter (*D*) and 1.8 mm height (*h*). In comparison, the second is a single layer cylindrical structure with *D* = 20 mm and *h* = 1.27 mm height dimensions. In addition, short-circuited pins were placed to make the antennas smaller, and gaps were added in the radiating elements [28]. In another study performed on rats in 2015, a three-layer helical antenna operating in the ISM band was designed with dimensions of *D* = 5.5 mm and *h* = 3.81 mm [29]. In particular, the literature studies in recent years show that the development of various antennas working in the human body is met with interest [30,31,32,33,34,35,36,37]. On the other hand, antenna designs that can be implanted in laboratory animals, such as rats are very limited, especially for preliminary laboratory experiments [26,27,28,29].

In this paper, a new compact modified two-arm rectangular spiral microstrip implantable antenna design that operates in rat skin for use in biosensing applications is introduced. The novelty of this paper is that the proposed antenna has a unique two-armed square spiral geometry along with a compact size of 10 × 10 × 2.56 mm^3^. Moreover, an important contribution of the paper is the evaluation of the radiation performances of the proposed antenna design such as S_11_, far field radiation pattern, and SAR simulations in a realistic homogeneous 3D rat model. The proposed implant antenna covers the ISM band (Industrial, Scientific, and Medical 2.4–2.48 GHz) determined by international organizations with 2.39–2.51 GHz and 4.8% bandwidth are provided [39,40,41]. To reduce the size of the proposed antenna, there is a short circuit pin in the structure, and a 50 Ω coaxial cable feeds the radiant element. The optimum design has been achieved through parametric studies on the metallic line around the two-armed rectangular spiral radiation element. In addition, in vitro measurements are made by preparing rat skin-mimicking liquid found in the literature and the input reflection coefficient results are presented.

## 2. Antenna Design

The proposed microstrip implant antenna (MIA) design for biosensing applications and its placement on rat skin are shown in Figure 1 and Figure 2, respectively. As seen in Figure 1, the MIA design consists of a two-arm rectangular spiral (TARS) with a surrounding metallic line between the dielectric superstrate and substrate (Rogers 3210, $\epsilon_{r}$ = 10.2, tanδ = 0.0027) supported by a ground plane.The TARS-MIA has a compact size of 10 × 10 × 2.56 mm^3^ and is excited by a 50 Ω coaxial feed line. In the simulation, we use “waveguide port” excitation. It is noted that the modeled coaxial feed structure here is based on a 50 Ω realistic Huber+Suhner SMA connector.

The operating frequency for a two-arm rectangular spiral is determined by the appropriate truncation in the arm lengths. The differential phase between two adjacent arms gradually deviates from 180° as the spiral arm lengthens. Significant radiation occurs somewhere along the spiral arm when the phase deviation gets close to 0°. In other words, at this point, the nearby currents are in phase. Thus, the antenna is in the active region where the radiation occurs as the spiral currents are unrestricted by each other and can propagate freely. The total length of the radiating spiral antenna element and the metallic line surrounding it and directly connected to the current path of the antenna determines the antenna’s resonant frequency. This length is designed to resonate in the ISM operating band of the proposed antenna. According to simplistic spiral geometry calculations, antenna radiation occurs where the spiral circumference is one wavelength. Accordingly, the total length of the proposed spiral antenna ($L_{total}$), i.e., the sum of the lengths of the TARS and the surrounding metallic line, is optimized to be approximately 130 mm ≃ 1.04$\;\times \;\lambda_{0}$, where $\lambda_{0}$ is the free space wavelength at 2.4 GHz.

Since rats are generally preferred as experimental animals in laboratory measurements, the numerical analysis of the proposed microstrip antenna is carried out in an electrical environment with rat skin characteristics ($\epsilon_{ms}\;(\omega )$, ρ = 1050 Kg/m^3^) within the scope of the modeling ($W_{f}$ × $L_{f}$ × $h_{f}$) in Figure 2.

In order to create a rat skin simulation environment, the two-pole Cole–Cole model coefficients in Table 1 are defined in the CST Microwave Studio program, and relevant numerical analyses are performed. The Cole–Cole model, also known as the Cole–Cole equation or the Cole–Cole plot, is a mathematical model developed by Arthur C. Cole and Robert H. Cole in the 1940s, used to explain the complex electrical behavior of materials [42]. The Cole–Cole model is used to describe the frequency-dependent behavior of materials, such as dielectric materials or biological tissues, which exhibit a distribution of relaxation times. In this model, the electrical properties of a material are characterized by a complex dielectric constant or impedance, which varies with frequency. The Cole–Cole equation describes the real and imaginary parts of the complex impedance as a function of frequency, and it can be used to fit experimental data to obtain the material’s electrical properties. The Cole–Cole model is a successful model that accurately expresses and represents biological tissues in a wide frequency range, and has been used in previous studies in the literature [26]. The Cole–Cole model can be expressed in terms of a complex permittivity, which relates the tissue’s response to an applied electric field. The complex permittivity is given by:(1)$$\epsilon^{\prime }-j\epsilon^{\prime \prime }=\epsilon_{\infty }+\sum_{n}\frac{\Delta \varepsilon_{n}}{1+{\left(j\omega \tau_{n}\right)}^{\left(1-\alpha_{n}\right)}}+\frac{\sigma_{i}}{j\omega \varepsilon_{0}},$$
where $\epsilon^{\prime }$ and $\epsilon^{\prime \prime }$ are relative dielectric permittivity and dielectric loss factor of the tissue as a function of frequency *w*, respectively. *n* is the order of the Cole–Cole model (with *n* = 1, 2, 3), $\epsilon_{\infty }$ is the permittivity at high frequencies (in the limit as *w* approaches infinity), $\Delta \varepsilon_{n}$ is the magnitude of the dispersion, $\tau_{n}$ is *n*. the degree relaxation time of the tissue in seconds, $\alpha_{n}$ is the parameter that allows for the broadening of the dispersion, and $\sigma_{i}$ is the static ionic conductivity in Siemens per meter (S/m).

The input reflection coefficient performances (S_11_) of the TARS MIA design in air and in rat skin (CST and HFSS) are given in Figure 3. As can be seen, the proposed antenna design provides S_11_ ≤ −10 dB performance in the ISM band in the frequency range of 2.39–2.51 GHz, with 4.8% bandwidth in the simulated rat skin environment, while it does not have any resonance in the air. The simulation results of the proposed antenna’s input impedance (Z_in_) are shown in Figure 4. Except for the level differences between the CST and HFSS input reflection coefficient simulation results, it is confirmed by Figure 3 that the operating frequencies are obtained in both software. On the other hand, the level differences between the S_11_ results are that the simulation programs have different networking approaches, they use different solver algorithms to solve the electromagnetic field equations, and the boundary condition approaches differ. After the S_11_ results are verified with two different software, subsequent simulations are performed using CST based on the finite integration technique (FIT) method. The reason for choosing the CST is its ease of use and high accuracy. We also note that CST simulations are performed using the time domain solver.

In addition to simple rat skin simulations, EM analysis of the proposed antenna in a 3D rat model is also performed. The 3D rat model used in the study is freely available in the “*.STL” file format, which is widely used in computer-aided design (CAD) and computer-aided engineering (CAE) in [43]. STL files represent the surface geometry of a 3D model as a collection of connected triangles or facets, with each triangle defined by three points in 3D space (vertices). The format does not include information about colors, textures, or other material properties. In this context, this 3D homogeneous rat model is imported to CST, and then the Cole–Cole equation coefficients are defined to the model so that the model can show the electrical properties of the rat skin. Thus, this 3D homogeneous rat model, depicted in Figure 5, is introduced into the CST simulation environment to observe the possible practical laboratory measurement performance of the proposed TARS antenna.

Then, the proposed antenna is placed in the model as seen in Figure 6. The antenna is positioned in the animal’s dorsal area to facilitate data exchange and 3 mm below the rat skin, as in a simplistic rat skin model in Figure 2. The electrical properties of the 3D rat model are the same as the rat skin Cole–Cole model coefficients previously defined in the simulations for Figure 2.

Figure 7 shows the input reflection coefficient simulation results of the TARS-MIA design in different rat skin model approaches. As can be seen, the relevant simulations are carried out on three models: simplistic, partial, and whole. In the simple model, the antenna is placed in the center of a 60 × 60 × 9.28 mm^3^ structure and the simulation boundaries are decided to cover the entire area. In the partial and the whole model, the TARS-MIA is placed in a 3D rat prototype as in Figure 6. In the partial and the whole model, the main difference is in the boundaries of the simulation environment, that is, the simulation environment is constrained to differ to cover some and all of the 3D rat model. The boundary conditions in CST depend on the type of excitation. The excitation here is based on a 50 Ω realistic Huber+Suhner SMA connector, and modeled as a “waveguide port” excitation. In “waveguide port” excitation, “open add space” should be used in all directions. Regarding the amount of space away from the antenna in any direction, by default CST sets quarter wavelength (at a center frequency of 2.5 GHz). While the default simulation boundaries of the CST are used in the simple and partial model, the boundaries are determined to cover the entire rat skin environment model. As can be seen in Figure 7, it is noteworthy that the input reflection coefficient simulation results of the proposed antenna in all models are quite similar except for some S_11_ level deviations.

In Figure 8a,b, the polar and 3D radiation pattern results of the ISM band TARS-MIA at 2.4 GHz is given, respectively. As can be seen, the proposed design has an unsymmetrical radiation characteristic in the E− and H− planes due to its miniature dimensions and high tissue loss/conductivity. The calculated directivity in the main beam direction at 2.4 GHz is 3.18 dBi. Moreover, the calculated efficiency and gain values are 0.13% and −18.41 dBi, respectively. It is known that the efficiency value is quite low (less than 1%) due to the high tissue loss and the miniature size of the antenna. Compared with similar studies in the literature [22,23,24], the efficiency of the proposed antenna is acceptable. Controlling the far field model is crucial for implant antennas as it directly affects the antenna’s performance and the overall system’s efficacy. In implantable medical devices, such as pacemakers, cochlear implants, or deep brain stimulators, the antennas are typically located within the human body and operate at frequencies that are absorbed and attenuated by human tissue. This can result in a significant reduction in the signal strength, which can impact the overall performance of the device. Accordingly, the inclusion of the concept of metasurfaces in the antenna structure as future studies may be useful to control the far-field model [44].

The 3D Specific Absorption Rate (SAR) results in different simulation environments resulting from applying a maximum power of 1W to the supply input of the proposed ISM band TARS-MIA antenna are given in Figure 9. According to the FCC (Federal Communications Commission) standard, the amount of power required to be applied to the antenna should be such that the SAR in the tissue is equal to less than 1.6 W/kg (1 g tissue) to prevent possible tissue degradation. Thus, the input power of the recommended antenna should not exceed the SAR standard, for simplistic, partial, and whole models, respectively, 2.93 mW, 3.89 mW, and 3.98 mW.

## 3. Effects of Critical Design Parameters on the Antenna Performance

During the optimization studies, it is observed that the $L_{3}$, $L_{2}$, and $L_{10}$ lengths in Figure 1 have significant effects on the ultimate radiation performance. Thus, in the proposed TARS-MIA design, these lengths ($L_{3}$, $L_{2}$, and $L_{10}$) are considered as critical design parameters in order to obtain the optimum radiation performance. In this context, the simulation results that evaluate the scalability of the TARS-MIA design and the possible effects of the related parameters on the fabrication-induced S_11_ performance will be given in this section, respectively.

We would also like to note that, like the above-mentioned critical lengths, the widths of the spiral lines are obtained as a result of a series of optimization. It has been observed that these widths have a slight effect on S_11_ performance throughout the optimization studies. Accordingly, using the available size effectively, the spiral widths are optimized as a result of a series of parametric analyses, with $W_{6}=0.6$ mm at the widest and $W_{3}=0.3$ mm at the narrowest. Thus, in the limited available size (*W* × *L*), the required electrical length of the spiral arms has been achieved for the antenna to radiate at the desired operating frequency ($f_{c}=2.4$ GHz, @ISM).

### 3.1. Effect of Length L_3_ on the Input Reflection Coefficient (S_11_)

The effect of the length of $L_{3}$ which belongs to the inner spiral geometry on the performance of ISM band S_11_ is given in Figure 10. As can be seen, when the length of $L_{3}$ is increased towards the optimum length $(L_{3}$ = 6 mm) in 1 mm intervals, the two bands formed around 2.4 GHz and 2.5 GHz are combined and the optimum TARS-MIA performance has been achieved with 4.8% bandwidth centered at 2.45 GHz.

### 3.2. Effect of Length L_2_ on the Input Reflection Coefficient (S_11_)

The effect of $L_{2}$ length of outer spiral geometry on ISM band S_11_ performance is given in Figure 11. As can be seen, when the length of $L_{2}$ is increased towards the optimum length ($L_{2}\;=\;9.6$ mm) in 1 mm intervals, the ISM band centered at 2.45 GHz is formed and plays a very critical role in the formation of the desired S_11_ performance.

### 3.3. Effect of Length L_10_ on the Input Reflection Coefficient (S_11_)

The effect of the length of $L_{10}$ on ISM band S_11_ performance is shown in Figure 12. As can be seen, there are two resonances centered at 2.4 GHz and 2.7 GHz, while the length of $L_{10}$ is absent. When it is increased towards the optimum length ($L_{10}=2$ mm) in 1 mm intervals, optimum TARS-MIA performance centered at 2.45 GHz is obtained. $L_{10}$ length stands out as another parameter that plays a critical role in the formation of the ISM band.

### 3.4. Effect of Superstrate Thickness *h* on Input Reflection Coefficient (S_11_)

As is well known, in implantable antennas, the superstrate plays a crucial role in determining the antenna’s radiation pattern and impedance matching and prevents direct contact between the implant antenna and the surrounding tissue. Therefore, the superstrate itself and its thickness are also critical design parameters in implantable antennas [22,23,24,38]. The primary importance of the superstrate is to improve the radiation efficiency of the implant antenna. Since the human body is a highly lossy medium for electromagnetic waves, it is difficult for the antenna to radiate efficiently without the presence of a superstrate. The superstrate helps to reduce the electromagnetic losses in the body, and allows the antenna to radiate more effectively. Moreover, the superstrate helps to shape the radiation pattern of the implant antenna. This is important in medical applications, where the implant antenna may need to communicate with an external device or provide therapy to a specific area of the body. Moreover, the superstrate can also help to match the impedance of the implant antenna to that of the surrounding tissue.

The superstrate can help to prevent direct contact between the implant antenna and the surrounding tissue. Since the antenna is implanted inside the body, it is essential to minimize any direct contact between the antenna and the tissue to prevent tissue damage, inflammation, and other adverse effects. Additionally, the superstrate can be designed to be biocompatible, which means that it is not harmful to the surrounding tissue and does not cause any adverse reactions in the body. Furthermore, the superstrate can also provide physical support to the implant antenna, helping to keep it in place and preventing any movement that may cause tissue damage or other adverse effects. In the study, the superstrate is the same as the antenna layer (Rogers 3210, $\epsilon_{r}$ = 10.2) and the thickness is set as *h* = 1.27, optimizing the antenna’s impedance to match the surrounding tissue impedance, which helps to reduce reflection losses and improve the overall performance of the implant antenna, as in Figure 13.

## 4. Results and Discussions

In order to observe the input reflection coefficient performance of ISM band TARS-MIA in the laboratory, the proposed ultimate antenna design was fabricated and measured in an artificial liquid that can mimic the electrical properties of rat skin. Figure 14 depicts the perspective (a), exploded (b), and detailed (c) photographs of fabricated ISM band TARS-MIA design which is obtained as a result of a series of parametric studies. For the fabricated MIA, Rogers 3210 (*h* = 1.27 mm, $\epsilon_{r}$ = 10.2) dielectric substrate was used. In addition, the photolithography fabrication technique was used to fabricate the prototype. Thanks to the precision of the photolithography technique in the order of micrometers (μm), the precision fabrication shown in Figure 14c is achieved. The ultimate ISM band TARS-MIA prototype in Figure 14b is obtained by combining the layers in Figure 14a and connecting them with a 50 Ω SMA connector. Moreover, it is noted that the S_11_ measurements were performed with the Rohde & Schwarz ZVB8 Vector Network Analyzer in an unisolated laboratory environment.

The input reflection coefficient (S_11_) measurement and simulation results of the proposed TARS-MIA in the air are shown in Figure 15. As can be seen, the measurement and simulation results of the prototype antenna in the air are compatible with each other, except for some frequency and level differences due to fabrication tolerances. In Figure 15, the (S_11_) magnitude ranges from 0 to −10 dB. The biggest difference in terms of level is observed around 3 GHz. While there are some differences in the (S_11_) levels, it is relatively more consistent in profile throughout the frequency. These differences are mainly due to fabrication tolerances and non-isolated measurement environment.

In addition, within the scope of the study, the mixture in Table 2 was prepared to obtain the actual performance of the fabricated TARS-MIA antenna in rat skin [26].

The input reflection coefficient (S_11_) measurement of the proposed TARS-MIA prototype antenna was performed in rat skin-mimicking liquid in an unisolated laboratory environment using the Rohde& Schwarz ZVB8 Vector Network Analyzer, as shown in Figure 16.

The S_11_ measurement and CST simulation results of the fabricated ISM band TARS-MIA in rat skin-mimicking liquid are given in Figure 17. As can be seen, although the measurement and simulation results of the fabricated ISM band TARS-MIA are compatible with each other in general profile, some discrepancies are observed in terms of bandwidth and S_11_ levels. According to the simulation results, 4.8% bandwidth is obtained in the 2.39–2.51 GHz range, while the prototype antenna achieved 4% bandwidth in the 2.4–2.5 GHz. In the measurements, we would also like to point out that the rat skin-mimicking liquid is insulated with an acrylic transparent coating used to protect the live and ground terminals of the connector from corrosion, and the outer surfaces of the connector that meet the skin mimic gel, in order to prevent erroneous measurements caused by short circuits with each other. These differences between S_11_ measurement and simulation results are due to differences between the ideal simulation and the approximate measurement environment, such as the fabrication and production tolerances of rat skin-mimicking liquid.

Finally, in order to reveal the contribution of the study to the current literature, the proposed TARS-MIA design is compared with the microstrip implant designs proposed in the literature, especially for use in experimental animals, and presented in Table 3 in terms of size and S_11_ performances. As can be seen, the proposed TARS-MIA design stands out with its miniature dimensions of 10 × 10 × 2.56 mm^3^ and S_11_ bandwidth of 4% in the ISM band. As a result, the proposed design, with its miniature size and reasonable radiation properties, is a good alternative that can be used in laboratory animals for biosensing applications.

## 5. Conclusions

In this study, a compact modified two-arm rectangular spiral microstrip implantable antenna design for ISM band biosensing applications is presented. Using the rat skin Cole–Cole model coefficients in the literature, the electrical environment of the rat skin is created in the simulations and the comparative CST and HFSS input reflection coefficient results of the antenna are given. The simulation result of the proposed antenna covers the frequency band of 2.39–2.51 GHz and offers 4.8% bandwidth. Moreover, polar and 3D far field radiation patterns of the antenna are given, and the directivity is 3.18 dBi. The maximum input power has been determined for different 3D rat models by paying attention to the SAR standard defined by international organizations. The optimum design of TARS-MIA has been achieved as a result of several parametric studies and its fabrication has been realized. Finally, the input reflection coefficient results of the prototype antenna in the prepared ISM band rat skin-mimicking liquid are measured by a network analyzer. The measurement of the TARS-MIA design covers the frequency range of 2.4–2.5 GHz and exhibits 4% bandwidth performance. Some frequency band shifts and level differences between the S_11_ measurement and the simulation results are considered to be due to differences between the ideal simulation and the approximate measurement environment, such as the fabrication and manufacturing tolerances of the rat skin-mimicking liquid. Moreover, these differences between real and simulated measurements observed in input reflection coefficient results may have resulted from measurements made in a non-isolated laboratory environment. In addition, the deviations in the amount of material used in the ISM band rat skin-mimicking liquid and the air gaps in the prepared liquid are factors affecting the input reflection coefficient performance. Considering its compact size and acceptable radiation performance, such as moderate directivity performance and acceptable SAR values, the proposed TARS-MIA design stands out as a very good alternative when compared to similar designs in the literature.

## Figures and Tables

**Figure 1 sensors-23-04883-f001:**
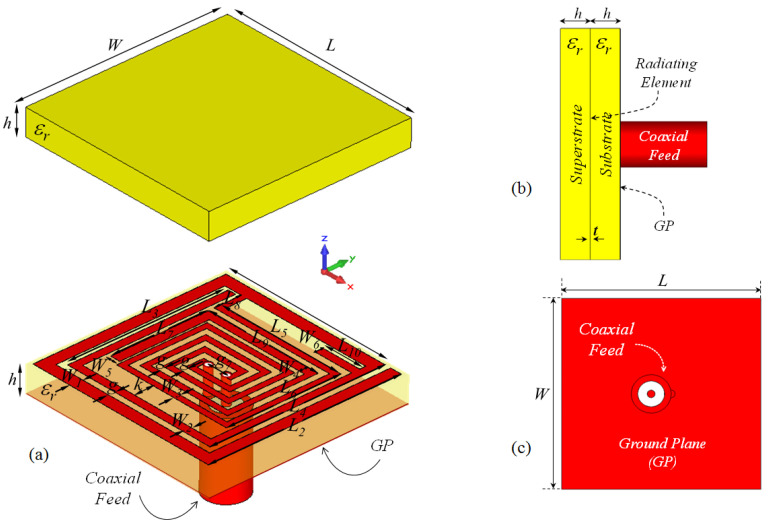
The proposed ISM band TARS-MIA configuration (**a**) exploded, (**b**) side, and (**c**) back views: *W* = *L* = 10, $L_{1}$ = 8.7, $L_{2}$ = 9.6, $L_{3}$ = 8, $L_{4}$ = 8.7, $L_{5}$ = 6.3, $L_{6}$ = 5.7, $L_{7}$ = 3.85, $L_{8}$ = 1.35, $L_{9}$ = 6, $L_{10}$ = 2, $W_{1}$ = 0.6, $W_{2}$ = $W_{4}$ = $W_{6}$ = 0.4, $W_{3}$ = 0.3, $W_{5}$ = 0.5, *k* = 0.95, *g* = 0.3, $g_{f}$ = 1, *h* = 1.27 (all in mm), $\epsilon_{r}$ = 10.2.

**Figure 2 sensors-23-04883-f002:**
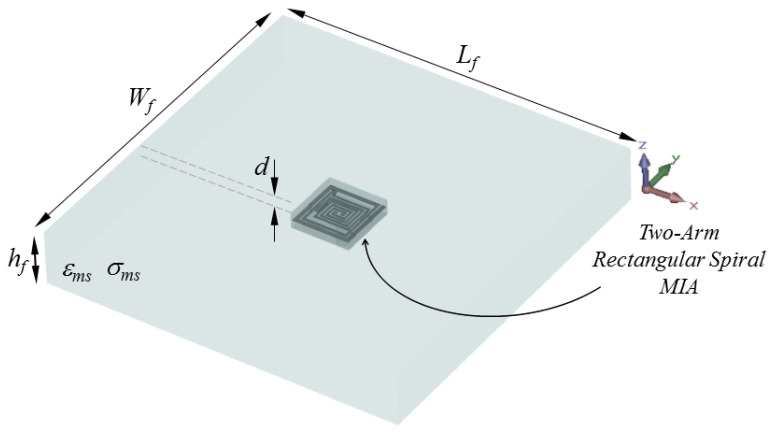
Placement of the proposed TARS-MIA Design in a simplistic rat skin model: $W_{f}$ = $L_{f}$ = 60, $h_{f}$ = 9.28, *d* = 3 (all in mm).

**Figure 3 sensors-23-04883-f003:**
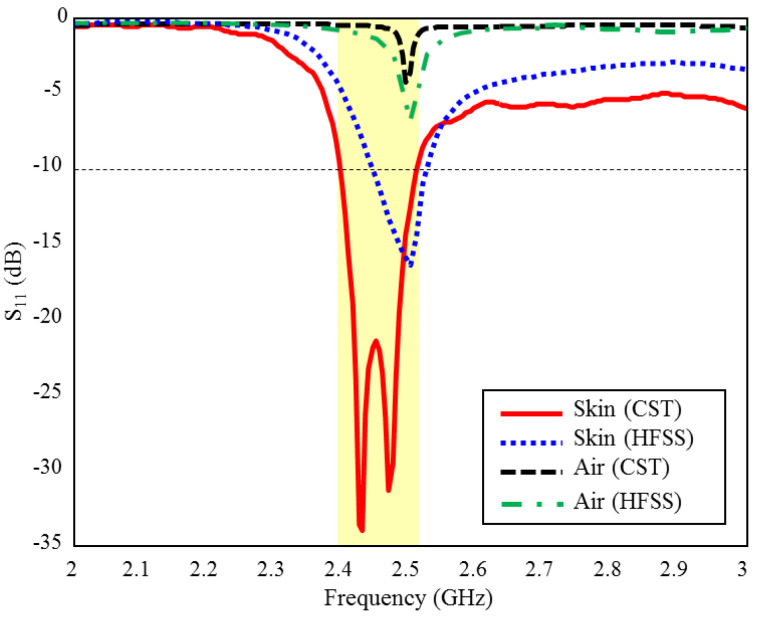
The input reflection coefficient (S_11_) simulation results of the ISM band TARS-MIA design in air and rat skin environment.

**Figure 4 sensors-23-04883-f004:**
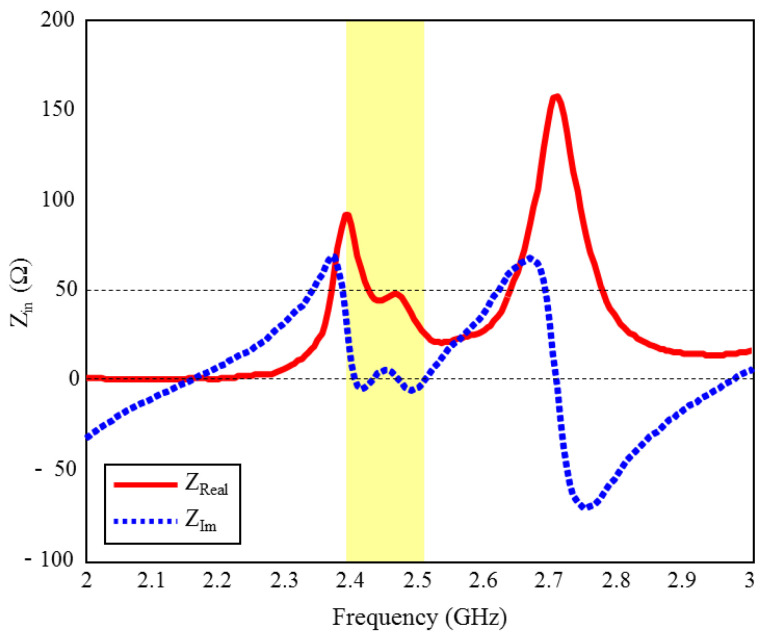
The input impedance (Z_in_) simulation results of the ISM band TARS-MIA design in rat skin environment.

**Figure 5 sensors-23-04883-f005:**
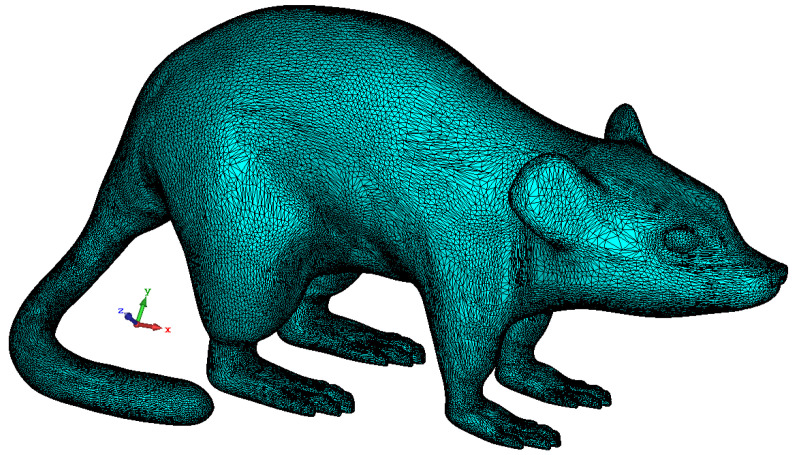
3D homogeneous simplistic rat model [43].

**Figure 6 sensors-23-04883-f006:**
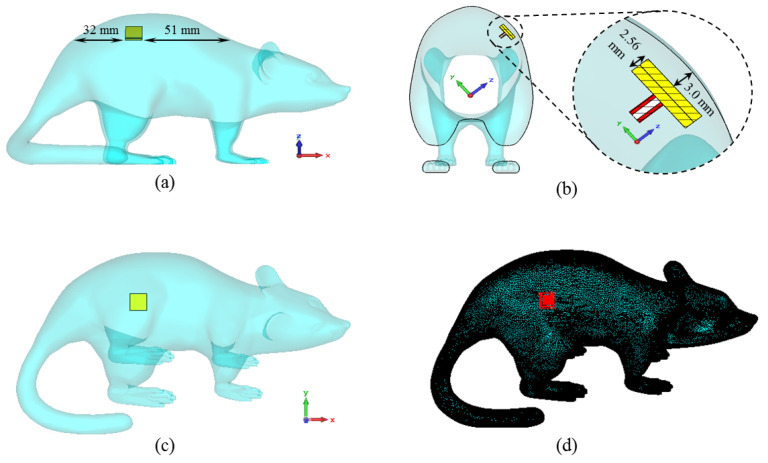
The placement of the ISM band TARS-MIA design along the x-axis (**a**), the z-axis (**b**), the y-axis (**c**), and the perspective view of the antenna through the model (**d**).

**Figure 7 sensors-23-04883-f007:**
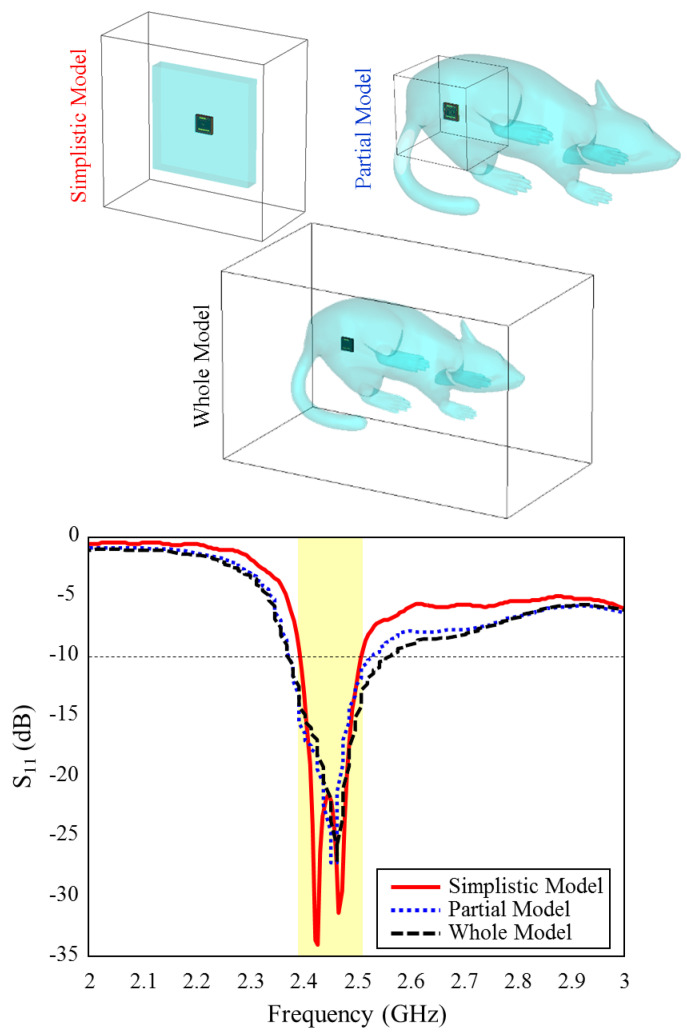
The input reflection coefficient results for different skin models and simulation boundaries of the TARS-MIA design.

**Figure 8 sensors-23-04883-f008:**
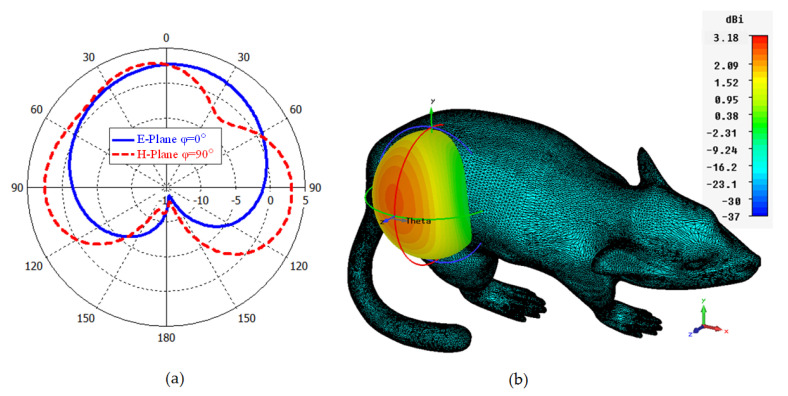
Polar (**a**) and 3D (**b**) far field radiation pattern results of the TARS-MIA design at 2.4 GHz.

**Figure 9 sensors-23-04883-f009:**
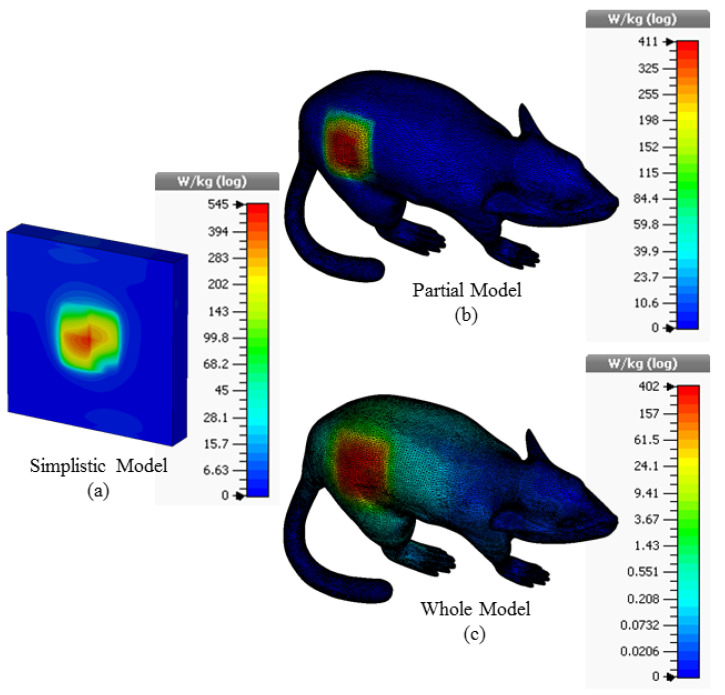
The SAR results for 1W input power value in the simplistic (**a**); partial (**b**); and whole (**c**) rat modeling approach for the TARS-MIA design.

**Figure 10 sensors-23-04883-f010:**
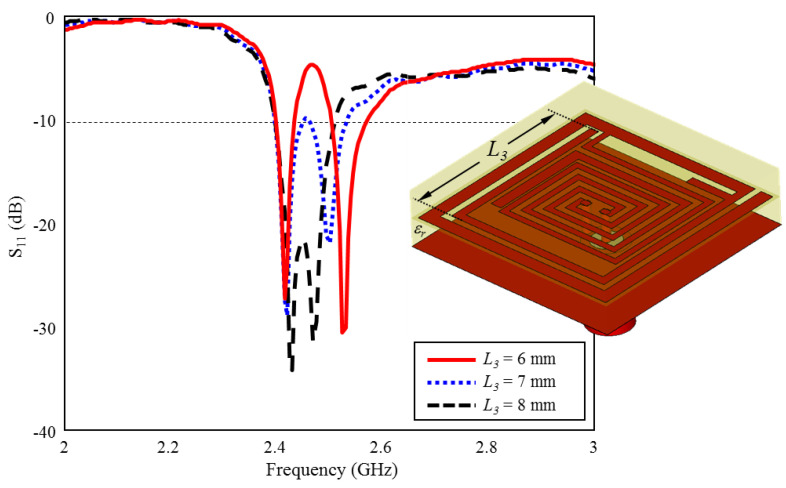
The effect of length $L_{3}$ on the input reflection coefficient (S_11_) performance.

**Figure 11 sensors-23-04883-f011:**
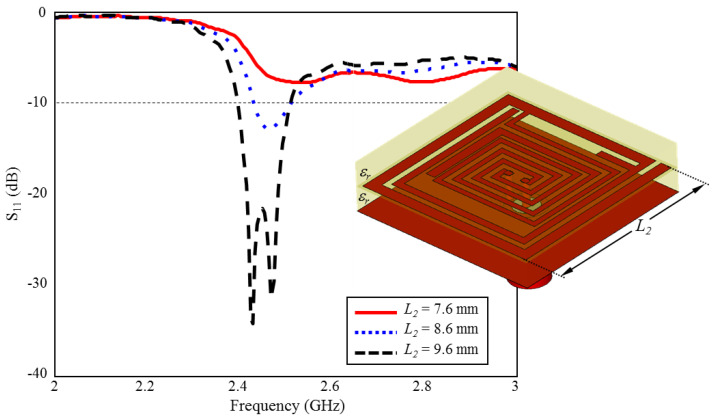
The effect of length $L_{2}$ on the input reflection coefficient (S_11_) performance.

**Figure 12 sensors-23-04883-f012:**
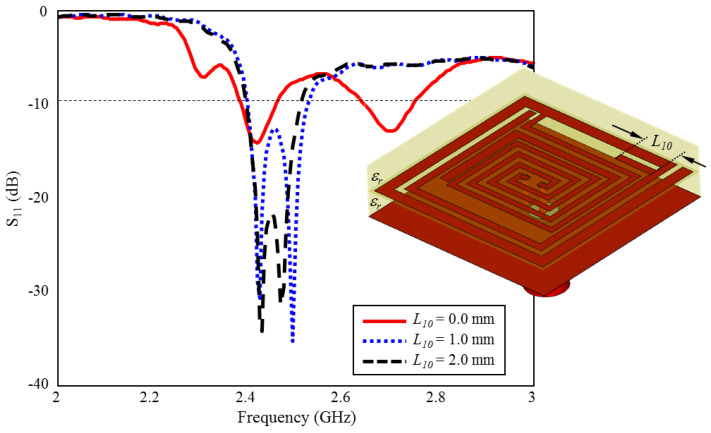
The effect of length $L_{10}$ on the input reflection coefficient (S_11_) performance.

**Figure 13 sensors-23-04883-f013:**
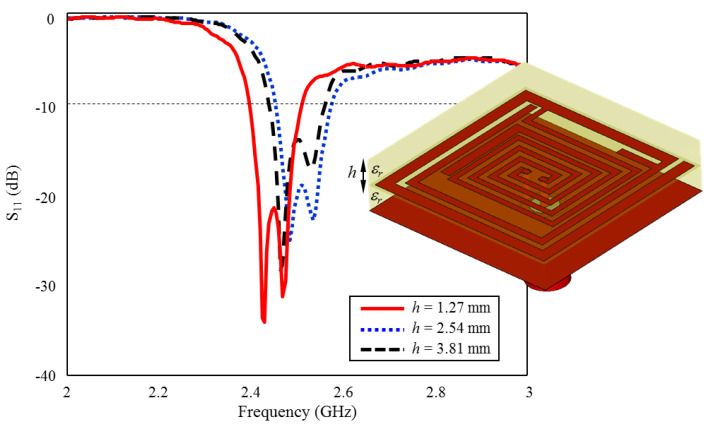
The effect of superstrate thickness *h* on the input reflection coefficient (S_11_) performance.

**Figure 14 sensors-23-04883-f014:**
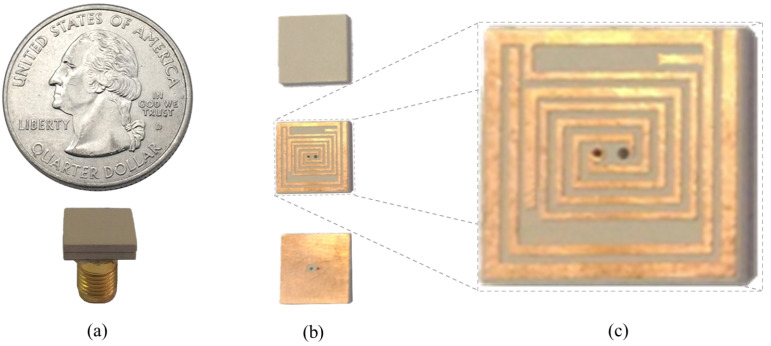
The perspective (**a**); exploded (**b**); and detailed (**c**) photographs of fabricated ISM band TARS-MIA.

**Figure 15 sensors-23-04883-f015:**
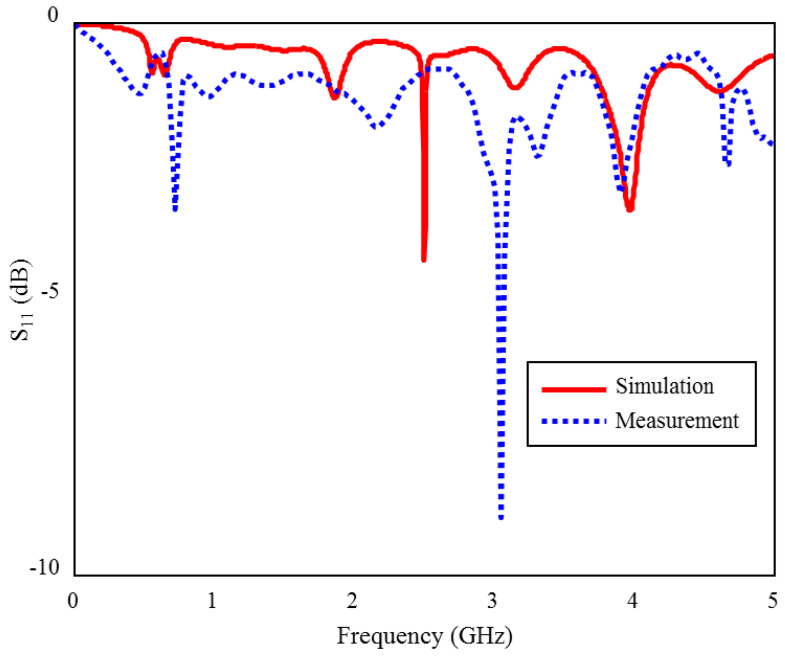
The S_11_ measurement and simulation results of the fabricated ISM band TARS-MIA design in air.

**Figure 16 sensors-23-04883-f016:**
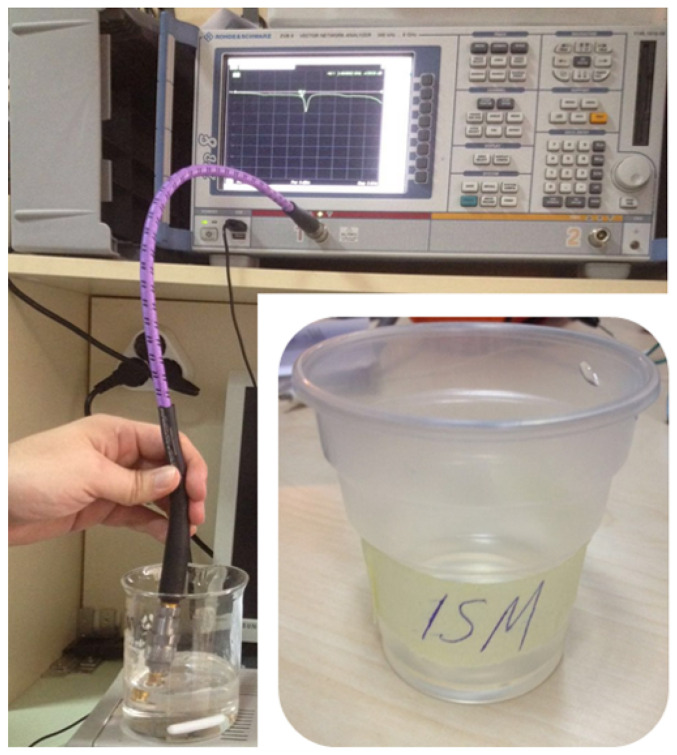
Measurement setup used to observe the input reflection coefficient performance (S_11_) of the TARS-MIA design in rat skin-mimicking liquid.

**Figure 17 sensors-23-04883-f017:**
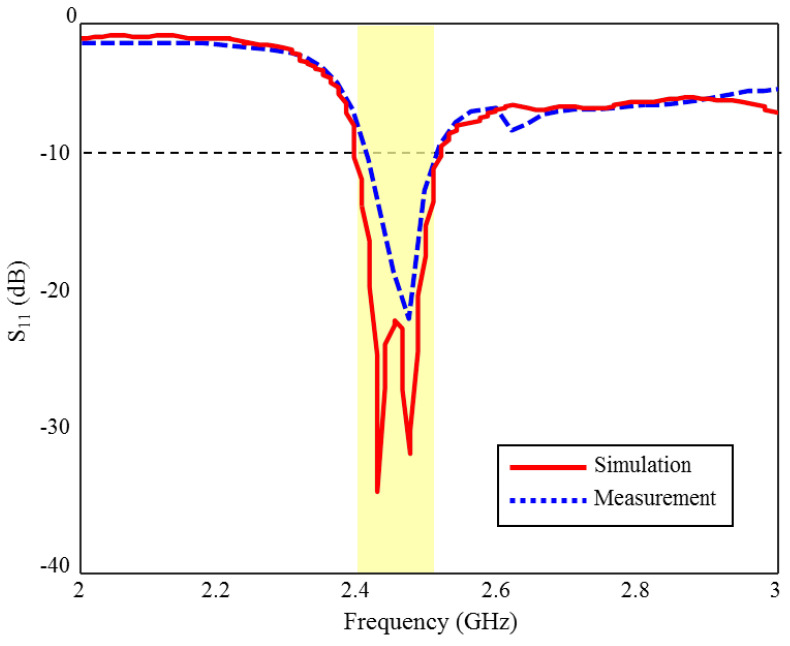
The S_11_ measurement and simulation results of the fabricated ISM band TARS-MIA design in rat skin-mimicking liquid.

**Table 1 sensors-23-04883-t001:** The Cole–Cole model coefficients for rat skin [26].

Coefficients	Single-Pole	Two-Pole	Three-Pole
$\varepsilon_{\infty }$	9.57	9.36	6.8
$\Delta \varepsilon_{1}$	21.17	19.76	21
$\tau_{1}\;\left(ps\right)$	11.32	10.02	7.6
$\Delta \varepsilon_{2}$	−	$2.9\times 10^{3}$	$2.8\times 10^{3}$
$\tau_{2}\;\left(ns\right)$	−	132.67	107.4
$\Delta \varepsilon_{3}$	−	−	$1.8\times 10^{6}$
$\tau_{3}\;\left(\mu s\right)$	−	−	215.7
$\sigma_{i}\;\left(S/m\right)$	1.28	1.38	1.33

**Table 2 sensors-23-04883-t002:** Rat Skin-mimicking Liquid Recipe for ISM Band [26].

Liquid Components	Mixture Percentage (%)
DGBE (Diethylene Glycol Butyl Ether)	6.98
Deionized Water	46.51
Triton X-100 (Polyethylene Glycol Monophenyl Ether)	46.51

**Table 3 sensors-23-04883-t003:** Comparison of the proposed implant antenna in the literature.

Ref.	Year	Tissue	Band	Bandwidth (MHz)	Geometry	Number of Layers	Dimensions (mm)	SC * Pin
[27]	2009	Rat Skin	MICS ISM	27 180	Meander Line	1	23 × 23 × 5	Yes
[29]	2015	Rat Muscle	ISM	355	Helical	3	D = 5.5, h = 3.81	No
[30]	2015	Human	MedRadio ISM	32 151	Meander Arms	1	27 × 9 × 1.27	Yes
[31]	2016	Human Skin	MICS ISM	30 168	Meander Strip	1	22 × 23 × 1.27	Yes
[33]	2018	Human Tissue	ISM	440	Four−Element	2	18.5 × 18.5 × 1.27	No
[34]	2018	Postmortem Human Subject	ISM	−	E-Shape with Meandering Arms	1	7.7 × 6.9 × 1.52	Yes
[35]	2019	Human Skin	ISM	184.1 219.7	Novel Flower−Shape	1	7 × 7.2 × 0.2	Yes
[36]	2019	Human Tissue	ISM	810	Conformal Patch	2	24 × 22 × 0.07	No
[37]	2020	Human Tissue	MICS ISM	127 903	Patch	2	D = 10, h = 2.54	Yes
	This Work	Rat Skin	ISM	100	Two-Arm rectangular spiral	2	10 × 10 × 2.54	Yes

* SC denotes to the “Short Circuit”.

## Data Availability

The original contributions presented in the study are included in the article, further inquiries can be directed to the corresponding author.

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
