# Peer review of "A Compact Modified Two-Arm Rectangular Spiral Implantable Antenna Design for ISM Band Biosensing Applications"

_sensors, 2023, doi:10.3390/s23104883_

Round 1

Reviewer 1 Report

The work in this paper is overall good. The paper is well-written. However, the following points must be taken care of.

1.       Line 108, it is better to explain, why or how it is decided to make the length of the surrounding metallic line equal to the length of the radiating element?

2.       In Figure 1, the four pictures should be denoted as a,b,c,d. In the bottom left picture, the white dimension annotations are not legible. Different color and larger font size might be better.

3.       In figure 1, the location of the shorting pin should be shown more clearly.

4.       Line 128 – 130, there might be some error in the notations.

5.       Line 137, the 3D rat model is not properly described. Is it a homogeneous model? Or inhomogeneous? Is the skin multilayered? This detail must be provided. Additionally, a reference of the model library should be added.

6.       Figure 4 legend should be Zim, instead of Zin.

7.       Figure 5, the location of the antenna along x-axis and z-axis is provided. But the location of the antenna along the y-axis is missing.

8.       Figure 5, why the antenna feeding port is pointing toward the inner portion of the animal? What specific implantable application is this antenna targeted for? The port should face outward if wireless power transfer is involved.

9.       Figure 6, what is the dimension of the simulation boundary in terms of wavelength? What boundary type is used?

10.    What software is used to obtain the plots in Figure 6? If CST, then whether time-domain or frequency-domain simulation is done? If HFSS, then isn’t the frequency range too large? What is the simulation frequency?

11.   Figure 6, the x-axis range should be narrower, such as 2 to 4 GHz. So that the bandwidth variation can be clearly compared among the three models.

12.   Line 187, the directivity of 3.18 dBi is obtained at what frequency?

13.   Line 162, should be rephrased as ‘ …..…. the amount of power required to be applied to the antenna should be such that the SAR in the tissue is equal to less than 1.6 W/kg  ….…… ’

14.   How will the antenna be excited inside the rat model? Will there be any wireless power transfer system? Was this system modeled in this work? It should be important to discuss and show the excitation technique of the antenna while placed inside the rat model. 

15.   The paper presents very few parametric analyses results. For antennas which utilize superstrate [1], the superstrate thickness is also a critical design parameter. This information is missing in this paper. It is better to include the analysis for the superstrate thickness, as in [1]. Or to the least, referring to this work [1] might suffice.

16.   How are the widths of the spiral lines designed? Were parametric analyses carried out to decide the line widths? A discussion must be added about the line widths.

17.   What is the shortest spacing between two adjacent lines? What antenna fabrication technique was used to fabricate the prototype? How small a spacing between adjacent copper lines can be etched using this method?  

18.   How was the feedline width designed?

19.   What tool/ method (wet etching, or milling machine or other?) was followed for fabrication?

20.   Figure 12a middle picture should be provided as a separate figure with high resolution.

21.   Figure 13, is this plot obtained with the superstrate or without the superstrate? What specific application is this antenna designed for? Is it only designed for rat skin?

22.   Figure 15, the x-axis range should be narrower, such as  2 to 4 GHz. So that the bandwidth variation can be clearly compared.

23.   Table 3, reference [28] and [32] should not be there, as these designed are in different frequency range. these works are not comparable with the proposed antenna. 

24.    Table 3, what does it mean by Number of Layers? The heading SC pin should be elaborate in the footnote. Beside the Bandwidth column, the center frequency should also be mentioned, otherwise it is not feasible to justify the comparison. Why the number of layers for the proposed antenna is mentioned to be 1?

25.   The strength of this work is that, the thin spacing between adjacent lines were fabricated and the measured s11 is reasonable. But the weakness of this work is the radiation pattern is not measured.

26.   What data are made available with the supplementary materials?

Reference:

[1] Parveen, Farhana, and Parveen Wahid. "Design of Miniaturized Antipodal Vivaldi Antennas for Wideband Microwave Imaging of the Head." Electronics 11.14 (2022): 2258.

Author Response

Dear Reviewer-1,

Thank you very much for the useful comments on the paper. We have addressed these recommendations and uploaded a revised version of our paper. Please find below (Please see the attachment) our responses to the Reviewer-1 and a summary of our revision.

Reviewer-1 (Reviewer(s)' Comments to Author)

Comments to the Author

The work in this paper is overall good. The paper is well-written. However, the following points must be taken care of.

  1. Line 108, it is better to explain, why or how it is decided to make the length of the surrounding metallic line equal to the length of the radiating element?

Thank you for your valuable suggestion. You are also very correct in your comment. In this paper, the proposed rectangular spiral antenna is extended to radiate in the ISM band, truncating the lengths of the rectangular spiral arms. The operation frequency for the two-arm rectangular spiral is established by this truncation. The differential phase between two adjacent arms gradually deviates from 180° as the spiral arm lengthens. Significant radiation occurs somewhere along the spiral arm when the phase deviation gets close to 0°. In other words, at this point, the nearby currents are in phase. Thus, the antenna is in the so-called "Active Region", where radiation occurs, as the spiral currents are unrestricted by one another and can freely radiate. The total length of the radiating spiral antenna element and the metallic line surrounding it and directly connected to the current path of the antenna determines the antenna's resonant frequency. This length is designed to resonate in the ISM operating band of the proposed antenna. According to simplistic spiral geometry calculations, antenna radiation occurs where the spiral circumference is one wavelength. That is, the spiral antenna proposed in this paper is optimized such that the overall length is approximately 130 mm =1.04λ, λ at ISM band.

In line with your suggestion, the following paragraph has been added to the relevant section.

The operating frequency for a two-arm rectangular spiral is determined by the appropriate truncation in the arm lengths. The differential phase between two adjacent arms gradually deviates from 180° as the spiral arm lengthens. Significant radiation occurs somewhere along the spiral arm when the phase deviation gets close to 0°. In other words, at this point, the nearby currents are in phase. Thus, the antenna is in the active region where the radiation occurs, as the spiral currents are unrestricted by each other and can propagate freely. The total length of the radiating spiral antenna element and the metallic line surrounding it and directly connected to the current path of the antenna determines the antenna's resonant frequency. This length is designed to resonate in the ISM operating band of the proposed antenna. According to simplistic spiral geometry calculations, antenna radiation occurs where the spiral circumference is one wavelength. Accordingly, the total length of the proposed spiral antenna, i.e. the sum of the lengths of the TARS and the surrounding metallic line, is optimized to be approximately 130 mm ≃ 1.04λ0, λ0 is the free space wavelength at 2.4 GHz.

  1. In Figure 1, the four pictures should be denoted as a,b,c,d. In the bottom left picture, the white dimension annotations are not legible. Different color and larger font size might be better.

Thank you for your valuable suggestion. Figure 1 has been updated in line with your recommendation.

  1. In Figure 1, the location of the shorting pin should be shown more clearly.

Thank you for your valuable suggestion. Figure 1 has been updated in line with your recommendation.

  1. Line 128 – 130, there might be some error in the notations.

Thank you for your careful review. We have meticulously corrected these overlooked typos. The paragraph in the relevant section has been completely rewritten and notations have been revised.

  1. Line 137, the 3D rat model is not properly described. Is it a homogeneous model? Or inhomogeneous? Is the skin multilayered? This detail must be provided. Additionally, a reference of the model library should be added.

Thank you for your valuable comment. The 3D rat model used in the study is freely available in the "*.STL" file format, which is widely used in the computer-aided design (CAD) and computer-aided engineering (CAE) at https://www.cadnav.com/. STL files represent the surface geometry of a 3D model as a collection of connected triangles or facets, with each triangle defined by three points in 3D space (vertices). The format does not include information about colors, textures, or other material properties. In this context, this 3D homogeneous rat model is imported to CST, and then the cole-cole equation coefficients are defined to the model so that the model can show the electrical properties of the rat skin. Thus, this simplistic homogeneous model is introduced to the CST simulation environment to observe the possible practical laboratory measurement performance of the proposed TARS antenna.

In line with your suggestion, the following paragraph has been added from line 151.

“In addition to simple rat skin simulations, EM analyze of the proposed antenna in a 3D rat model is also performed. The 3D rat model used in the study is freely available in the "*.STL" file format, which is widely used in the computer-aided design (CAD) and computer-aided engineering (CAE) at https://www.cadnav.com/. STL files represent the surface geometry of a 3D model as a collection of connected triangles or facets, with each triangle defined by three points in 3D space (vertices). The format does not include information about colors, textures, or other material properties. In this context, this 3D homogeneous rat model is imported to CST, and then the cole-cole equation coefficients are defined to the model so that the model can show the electrical properties of the rat skin. Thus, this 3D homogeneous rat model, depicted in Figure 5 is introduced into the CST simulation environment to observe possible practical laboratory measurement performance of the proposed TARS antenna.”

And the following reference has been cited in the paper by adding it to the references section.

[42] CadNav. Laboratory rat 3D Model@ONLINE. https://www.cadnav.com/3d-models/model-19550.html, 2023. [Online; accessed 382 20-April-2023]

  1. Figure 4 legend should be Zim, instead of Zin.

Thank you for your careful review. Figure 4 legend has been updated as ZIm line with your recommendation.

  1. Figure 5, the location of the antenna along x-axis and z-axis is provided. But the location of the antenna along the y-axis is missing.

Thank you for your valuable suggestion. Figure 5 has been updated by providing the location of the antenna along the y-axis.

  1. Figure 5, why the antenna feeding port is pointing toward the inner portion of the animal? What specific implantable application is this antenna targeted for? The port should face outward if wireless power transfer is involved.

The reason why the antenna feeding port is pointing toward the inner portion of the animal is because the antenna's maximum radiation direction is along the z-axis. The proposed antenna is targeted for sensing application where in-body health data will be transferred.

  1. Figure 6, what is the dimension of the simulation boundary in terms of wavelength? What boundary type is used?

The boundary conditions in CST depend on the type of excitation. In simulation we use   "waveguide port" excitation. The modeled excitation here is based on 50 ohm realistic Huber+Suhner SMA connector. In "waveguide port" excitation, it should be used "open add space" in all directions. Regarding the amount of space away from the antenna in any direction, by default CST sets quarter wavelength (at center frequency of 2.5 GHz), but you can edit this and increase to half wavelength or full wavelength, but it will be in trade-off simulation time. While the default simulation boundaries of the CST are used in the simple and partial model, the boundaries are determined to cover the entire rat skin environment model.

In line with your comment and the assessment made above, the following paragraph has been added from line 189.

“The boundary conditions in CST depend on the type of excitation. The excitation here is based on 50 ohm realistic Huber+Suhner SMA connector, and modelled as a "waveguide port" excitation. In "waveguide port" excitation, it should be used "open add space" in all directions. Regarding the amount of space away from the antenna in any direction, by default CST sets quarter wavelength (at center frequency of 2.5 GHz). While the default simulation boundaries of the CST are used in the simple and partial model, the boundaries are determined to cover the entire rat skin environment model”

  1. What software is used to obtain the plots in Figure 6? If CST, then whether time-domain or frequency-domain simulation is done? If HFSS, then isn’t the frequency range too large? What is the simulation frequency?

CST software is used to obtain the plots in Figure 6. Time domain simulation is also performed in CST. As the simulation frequency, 2.4 GHz, which is almost the center of the ISM band, is taken.

In line with your comment and the assessment made above, the following paragraph has been added from line 193.

“Except for the level differences between the CST and HFSS input reflection coefficient simulation results, it is confirmed by Figure 3 that the operating frequencies are obtained in both software. On the other hand, the level differences between the S11 results are that the simulation programs have different networking approaches, they use different solver algorithms to solve the electromagnetic field equations, and the boundary condition approaches differ.

After the S11 results were verified with two different software, subsequent simulations were performed using CST based on the finite integration technique (FIT) method. The reason for choosing the CST is its ease of use and high accuracy.  Also, we note that CST simulations are performed using time domain solver.”

  1. Figure 6, the x-axis range should be narrower, such as 2 to 4 GHz. So that the bandwidth variation can be clearly compared among the three models.

Formerly Figure 6, newly Figure 7, updated in line with your suggestion.

  1. Line 187, the directivity of 3.18 dBi is obtained at what frequency?

The directivity of 3.18 dBi is obtained at the 2.4 GHz frequency.

  1. Line 162, should be rephrased as ‘ …..…. the amount of power required to be applied to the antenna should be such that the SAR in the tissue is equal to less than 1.6 W/kg ….…… ’

Thank you for your valuable suggestion. In line with your suggestion, line 188 is rephrased as “According to the FCC (Federal Communications Commission) standard, the amount of power required to be applied to the antenna should be such that the SAR in the tissue is equal to less than 1.6 W/kg (1g tissue) to prevent possible tissue degradation."

  1. How will the antenna be excited inside the rat model? Will there be any wireless power transfer system? Was this system modeled in this work? It should be important to discuss and show the excitation technique of the antenna while placed inside the rat model.

The antenna proposed in this study is a prototype design developed for an in-body health data transfer system. The in-body health system that is planned to be developed is a microcomputer-assisted sensor-based system that is planned to be an internal power source. And it is currently under development. At this stage, the feeding method, which is included in the entire system, is not part of this study. Our work still continues.

  1. The paper presents very few parametric analyses results. For antennas which utilize superstrate [1], the superstrate thickness is also a critical design parameter. This information is missing in this paper. It is better to include the analysis for the superstrate thickness, as in [1]. Or to the least, referring to this work [1] might suffice.

Thank you for your valuable suggestion. The superstrate is a dielectric layer that is placed on top of the implant antenna, and it helps to control the electromagnetic fields around the antenna. As is well known, in implantable antennas, the superstrate plays a crucial role in determining the antenna's radiation pattern, impedance matching, and prevent direct contact between the implant antenna and the surrounding tissue. So, the superstrate itself and its thickness are also critical design parameters in implantable antennas. The primary importance of the superstrate is to improve the radiation efficiency of the implant antenna. Since the human body is a highly lossy medium for electromagnetic waves, it is difficult for the antenna to radiate efficiently without the presence of a superstrate. The superstrate helps to reduce the electromagnetic losses in the body and allows the antenna to radiate more effectively. Also, the superstrate also helps to shape the radiation pattern of the implant antenna. This is important in medical applications, where the implant antenna may need to communicate with an external device or provide therapy to a specific area of the body. Moreover, the superstrate can also help to match the impedance of the implant antenna to that of the surrounding tissue.

The superstrate can also help to prevent direct contact between the implant antenna and the surrounding tissue. Since the antenna is implanted inside the body, it is essential to minimize any direct contact between the antenna and the tissue to prevent tissue damage, inflammation, and other adverse effects. Additionally, the superstrate can be designed to be biocompatible, which means that it is not harmful to the surrounding tissue and does not cause any adverse reactions in the body. Furthermore, the superstrate can also provide physical support to the implant antenna, helping to keep it in place and preventing any movement that may cause tissue damage or other adverse effects. In the study, the superstrate is the same as the antenna layer (Rogers 3210, εr = 10.2) and the thickness is set as h=1.27, optimizing the antenna's impedance to match the surrounding tissue impedance, which helps to reduce reflection losses and improve the overall performance of the implant antenna, as in Figure 12.

In line with your suggestion, the above information and the below Figure (namely Figure 13) are included and the paper you suggested is cited as [38].

[38] Parveen, F.; Wahid, P. Design of Miniaturized Antipodal Vivaldi Antennas for Wideband Microwave Imaging of the Head. 442

Electronics 2022, 11, 2258.

  1. How are the widths of the spiral lines designed? Were parametric analyses carried out to decide the line widths? A discussion must be added about the line widths.

Like all dimensions of the ultimate antenna design in the paper, the widths of the spiral lines were obtained as a result of optimization study. It has been observed that these widths have slight effect on S11 performance throughout the optimization studies. Accordingly, using the available size effectively, the spiral widths were optimized as a result of a series of parametric analyses, with W6=0.6 mm at the widest and W3=0.3 mm at the narrowest. Thus, in the limited available size (W×L), the required electrical length of the spiral arms has been achieved for the antenna to radiate at the desired operating frequency (fc=2.4 GHz, @ISM)”

In line with your suggestion, the following paragraph has been included in the paper starting from line 203.

“We would also like to note that, like the above-mentioned critical lengths, the widths of the spiral lines were obtained as a result of a series of optimization. It has been observed that these widths have slight effect on S11 performance throughout the optimization studies. Accordingly, using the available size effectively, the spiral widths were optimized as a result of a series of parametric analyses, with W6=0.6 mm at the widest and W3=0.3 mm at the narrowest. Thus, in the limited available size (W×L), the required electrical length of the spiral arms has been achieved for the antenna to radiate at the desired operating frequency (fc=2.4 GHz, @ISM).”

  1. What is the shortest spacing between two adjacent lines? What antenna fabrication technique was used to fabricate the prototype? How small a spacing between adjacent copper lines can be etched using this method?

The shortest spacing between two adjacent lines is 0.1 mm. Photolithography fabrication technique was used to fabricate the prototype. The precision of photolithography can range from a few micrometers (μm) to a few tens of micrometers.

Based on your comment, the following explanation is included in the "Results and Discussions" Section.

In addition, the photolithography fabrication technique was used to fabricate the prototype. Thanks to the precision of the photolithography technique in the order of micrometers (μm), the precision production shown in Figure 14(c) is achieved.”

  1. How was the feedline width designed?

The feed line is modeled using the macro included in the CST simulation program (screenshot is below). We would also like to point out that the modeled excitation model here is a 50 ohm compatible Huber+Suhner SMA connector.

  1. What tool/ method (wet etching, or milling machine or other?) was followed for fabrication?

Photolithography fabrication technique was used to fabricate the prototype. Techniques such as wet etching or milling machine were not used.

  1. Figure 12a middle picture should be provided as a separate figure with high resolution.

In line with your suggestion, former Figure 12a, new Figure 14, the antenna plane photo is provided as a separate high resolution figure.

  1. Figure 13, is this plot obtained with the superstrate or without the superstrate? What specific application is this antenna designed for? Is it only designed for rat skin?

This graph was obtained with the superstrate. This implantable antenna is developed for in-body biosensing applications. It is designed for rat skin because it is intended for use in pioneering laboratory experiments. Afterwards, it will be optimize for human skin implementation.

  1. Figure 15, the x-axis range should be narrower, such as 2 to 4 GHz. So that the bandwidth variation can be clearly compared.

Based on your suggestion, Figure 15 (now, Figure 17) is replotted with a narrower x-axis range of 2 to 3 GHz.

  1. Table 3, reference [28] and [32] should not be there, as these designed are in different frequency range. these works are not comparable with the proposed antenna.

Thank you for your valuable suggestion. The references you mentioned have been removed from the table.

  1. Table 3, what does it mean by Number of Layers? The heading SC pin should be elaborate in the footnote. Beside the Bandwidth column, the center frequency should also be mentioned, otherwise it is not feasible to justify the comparison. Why the number of layers for the proposed antenna is mentioned to be 1?

The Number of Layers corresponds to the number of dielectric layers used. There are dual, triple and multi-layer designs in the literature. Considering the volumetric comparison of the designs, the number of layers used in the table is also indicated.

In line with your suggestion, the SC pin heading is elaborate in the footnote. Since there is already a heading for the operating band (ISM/ MICS) and the table is already quite large, there is no need to add a separate header for the center frequency. If you insist on this assessment, we can update the table.

Thank you for your careful review. The number of layers for the proposed antenna was incorrectly specified as 1. This number has been corrected as 2 in the table.

  1. The strength of this work is that, the thin spacing between adjacent lines were fabricated and the measured s11 is reasonable. But the weakness of this work is the radiation pattern is not measured

Thank you for your comment. Unfortunately, we could not perform the radiation pattern measurements, since the non-isolated laboratory environment required for the measurement of the radiation pattern, especially for the measurement in liquid, is not available.

  1. What data are made available with the supplementary materials?

No data is presented in this study along with any Supplementary materials. The title of "Data Availability Statement" in the paper has been revised accordingly.

Reviewer 2 Report

1. The article is within the scope of the journal. The authors presented a well-organized paper and easy-to-read and to-follow.

2. In this paper, the authors have proposed the design of a microstrip implantable antenna based on the two-arm rectangular spiral element for ISM band biotelemetric sensing applications.

3. The radiating element consists of a two-arm rectangular spiral placed on a ground-supported dielectric layer. A superstrate of the same material is used to prevent the contact of tissue and metallic radiator element.

4. To validate the theoretical results of the proposed antenna, the authors have carried out measurements using a prototype of the fabricated antenna. The in-vitro measurement and simulation results were compatible, except for some inconsistencies due to manufacturing and material tolerances.

5. The authors have compared the performance of the proposed antenna with those published in the literature to reveal the robustness of the proposed one.

6. There is a clear methodology. There is an extensive explanation of the method and a discussion of the results.

However, I suggest the authors address the following issues:

1. Return loss in dB is positive, and the reflection coefficient is negative. Please, take care of the terms and their physical meaning. The term (Return loss) should be replaced, wherever it is mentioned in the text, by either (input reflection coefficient) or only (S11). For more information, the authors are advised to see:

T. S. Bird, "Definition and Misuse of Return Loss [Report of the Transactions Editor-in-Chief]," in IEEE Antennas and Propagation Magazine, vol. 51, no. 2, pp. 166-167, April 2009.

 doi: 10.1109/MAP.2009.5162049

2. The language of the article has to be slightly revised. The attached file contains many suggestions and comments on the first three pages.

Author Response

Dear Reviewer-2,

Thank you very much for the useful comments on the paper. We have addressed these recommendations and uploaded a revised version of our paper. Please find below (Please see the attachment) our responses to the Reviewer-1 and a summary of our revision.

Reviewer-2 (Reviewer(s)' Comments to Author)

Comments to the Author

The work in this paper is overall good. The paper is well-written. However, the following points must be taken care of. The article is within the scope of the journal. The authors presented a well-organized paper and easy-to-read and to-follow. In this paper, the authors have proposed the design of a microstrip implantable antenna based on the two-arm rectangular spiral element for ISM band biotelemetric sensing applications. The radiating element consists of a two-arm rectangular spiral placed on a ground-supported dielectric layer. A superstrate of the same material is used to prevent the contact of tissue and metallic radiator element. To validate the theoretical results of the proposed antenna, the authors have carried out measurements using a prototype of the fabricated antenna. The in-vitro measurement and simulation results were compatible, except for some inconsistencies due to manufacturing and material tolerances. The authors have compared the performance of the proposed antenna with those published in the literature to reveal the robustness of the proposed one. There is a clear methodology. There is an extensive explanation of the method and a discussion of the results.

Thank you for your valuable comments and evaluations. We appreciate the valuable time you have devoted to review the paper.

However, I suggest the authors address the following issues:

  1. Return loss in dB is positive, and the reflection coefficient is negative. Please, take care of the terms and their physical meaning. The term (Return loss) should be replaced, wherever it is mentioned in the text, by either (input reflection coefficient) or only (S11). For more information, the authors are advised to see:
  2. S. Bird, "Definition and Misuse of Return Loss [Report of the Transactions Editor-in-Chief]," in IEEE Antennas and Propagation Magazine, vol. 51, no. 2, pp. 166-167, April 2009. doi: 10.1109/MAP.2009.5162049.

Thank you for your kind notice. This is actually a common misuse in various articles and books published by many authors. This misuse is expressed clearly in the report that you mentioned "Definition and Misuse of Return Loss [Report of the Transactions Editor-in-Chief]," published in 2009 by Prof Bird, who is the editor-in-chief of the IEEE TAP magazine at that time. As you stated, Return Loss is basically ratio between Incident power (Pi) on load to Reflected power (Pr) back to source. So Return Loss is positive quantity RL (dB)= Incident power ( Pi)/Reflected power (Pr), RL(dB)= 10log(Pi/Pr). Scattering parameter or S11 is basically ratio between Reflected power (Pr) to Incident power (Pi). So Scattering parameter or S11 is negative quantity S11 (dB) = 10log(Pr/ Pi). In line with your suggestion, the expression "Return Loss" in the paper has been changed to “input reflection coefficient" or only “|S11|".

  1. The language of the article has to be slightly revised. The attached file contains many suggestions and comments on the first three pages

Thank you for your correction suggestions. The entire paper has been thoroughly revised for grammatical and typos. Corrections are highlighted in red font.

Reviewer 3 Report

In this paper, Mustafa et al. presents a new microstrip implantable antenna (MIA) design based on the 1 two-arm rectangular spiral (TARS) element for ISM band (Industrial, Scientific and Medical 2.4−2.48 2 GHz) biotelemetric sensing applications. The radiating element consists of a two-arm rectangular spiral placed on a ground supported dielectric layer and a metallic line surrounding this spiral, and superstrate is used to prevent the contact of tissue and 6 metallic radiator element with each other. The impedance bandwidth of the TARS-MIA is from 2.39 to 2.51 GHz considering 50 Ω system and has directional radiation pattern wtih directivity of − 3.18 dBi. The in-vitro return loss measurements are realized in a rat skin mimicking liquid 13 reported in literature. It was observed that the in-vitro measurement and simulation results were quite compatible with each other, the proposed TARS-MIA can be a good alternative for 16 ISM-band biotelemetry operations with its small size and acceptable radiation performance. The paper in whole is well organized, and here are some minor comments:

(1) The simulation results are different with CST and HFSS as shown in Fig. 3. So how to select the simulation software can be briefly discussed.

(2) It seems there are some difference between the simulation and measurements in Fig. 13. It should be extensively discussed.

(3) The simulation results with superstrate should be added, since the relative permittivity will affect the performance of the antenna, both return loss and far field pattern.

(4) The concept of metasurfaces might be helpful for control the farfield pattern, for example DOI: 10.1002/lpor.202200777.

Author Response

Dear Reviewer-3,

Thank you very much for the useful comments on the paper. We have addressed these recommendations and uploaded a revised version of our paper. Please find below (Please see the attachment) our responses to the Reviewer-3 and a summary of our revision.

Reviewer-3 (Reviewer(s)' Comments to Author)

Comments to the Author

In this paper, Mustafa et al. presents a new microstrip implantable antenna (MIA) design based on the 1 two-arm rectangular spiral (TARS) element for ISM band (Industrial, Scientific and Medical 2.4−2.48 2 GHz) biotelemetric sensing applications. The radiating element consists of a two-arm rectangular spiral placed on a ground supported dielectric layer and a metallic line surrounding this spiral, and superstrate is used to prevent the contact of tissue and 6 metallic radiator element with each other. The impedance bandwidth of the TARS-MIA is from 2.39 to 2.51 GHz considering 50 Ω system and has directional radiation pattern wtih directivity of − 3.18 dBi. The in-vitro return loss measurements are realized in a rat skin mimicking liquid 13 reported in literature. It was observed that the in-vitro measurement and simulation results were quite compatible with each other, the proposed TARS-MIA can be a good alternative for 16 ISM-band biotelemetry operations with its small size and acceptable radiation performance. The paper in whole is well organized, and here are some minor comments:

Thank you for your valuable comments and evaluations. We appreciate the valuable time you have devoted to review the paper.

  • The simulation results are different with CST and HFSS as shown in Fig. 3. So how to select the simulation software can be briefly discussed.

Selecting the appropriate EM simulation software can be a challenging task. There are many factors to consider, such as the type of problem, the accuracy required, the size of the problem, the complexity of the geometry, the computational resources available, and the budget.  CST and HFSS have different strengths and weaknesses depending on the application domain. CST is typically used for high-frequency applications such as antennas, microwave circuits, and RF components, while HFSS is more commonly used for solving electromagnetic problems in the GHz range and above, such as waveguide components, microwave filters, and electromagnetic compatibility (EMC) analysis. The accuracy of the simulation results depends on how well the geometry is represented and how finely the mesh is constructed. CST and HFSS have different approaches to meshing, with CST using a tetrahedral mesh and HFSS using a hybrid mesh. Depending on the geometry and the simulation requirements, one approach may be more suitable than the other. In addition, both CST and HFSS use different solver algorithms to solve the electromagnetic field equations. CST uses a finite integration technique (FIT) solver, while HFSS uses a finite element method (FEM) solver. Each solver has its own advantages and disadvantages depending on the problem at hand. In fact, the main reason we chose the CST in the paper is that we already have a new version and it converges to the results using Transient Solver with high accuracy. HFSS is used to validate preliminary results at the beginning of the study. As the CST and HFSS results were almost close to each other (at least the operation bands could be observed in both simulations), it was concluded that we were on the right track, and CST is used in subsequent studies.

In line with your comment and the assessment made above, the following paragraph has been added from line 193.

“Except for the level differences between the CST and HFSS input reflection coefficient simulation results, it is confirmed by Figure 3 that the operating frequencies are obtained in both software. On the other hand, the level differences between the S11 results are that the simulation programs have different networking approaches, they use different solver algorithms to solve the electromagnetic field equations, and the boundary condition approaches differ.

After the S11 results were verified with two different software, subsequent simulations were performed using CST based on the finite integration technique (FIT) method. The reason for choosing the CST is its ease of use and high accuracy.  Also, we note that CST simulations are performed using time domain solver.”

(2) It seems there are some difference between the simulation and measurements in Fig. 13. It should be extensively discussed.

In Figure 13, the S11 magnitude axis ranges from 0 to -10 dB. The biggest difference in terms of level is observed around 3 GHz. Although there are some differences in level, it is relatively more compatible in profile along the frequency axis. These differences are mainly due to fabrication tolerances and non-isolated measurement environment.

 In line with your suggestion, the following paragraph is included in the relevant section.

“In Figure 13, the S11 magnitude axis ranges from 0 to -10 dB. The biggest difference in terms of level is observed around 3 GHz. While there are some differences in the S11 levels, it is relatively more consistent in profile throughout the frequency. These differences are mainly due to fabrication tolerances and non-isolated measurement environment.”

(3) The simulation results with superstrate should be added, since the relative permittivity will affect the performance of the antenna, both return loss and far field pattern.

Superstrate is used in all simulations in the paper. All results in the study are results with superstrate.

(4) The concept of metasurfaces might be helpful for control the farfield pattern, for example DOI: 10.1002/lpor.202200777.

Thank you for your suggestion. In line with your suggestion, it is stated with the following paragraph included in the relevant section that the control of the far field radiation diagram with the metasurfaces concept can be achieved within the scope of the future study.

“Controlling the far field model is crucial for implant antennas as it directly affects the antenna's performance and the overall system's efficacy. In implantable medical devices, such as pacemakers, cochlear implants, or deep brain stimulators, the antennas are typically located within the human body and operate at frequencies that are absorbed and attenuated by human tissue. This can result in a significant reduction in the signal strength, which can impact the overall performance of the device. Accordingly, inclusion of the concept of metasurfaces in the antenna structure as future studies may be useful to control the far-field model”

Reviewer 4 Report

1.    Key words should be in alphabetical order.

2.    Abstract and conclusion should be concise.

Author Response

Dear Reviewer-4,

Thank you very much for the useful comments on the paper. We have addressed these recommendations and uploaded a revised version of our paper. Please find below (Please see the attachment) our responses to the Reviewer-4 and a summary of our revision.

  1. Key words should be in alphabetical order.

In line with your suggestion, the keywords have been rearranged in alphabetical order. 

  1. Abstract and conclusion should be concise.

In line with your suggestion, the entire article, as well as the abstract and conclusion sections have been revised.

Reviewer 5 Report

Dear Authors:

1- The article is well written and organized.

2- The abstract reflects the contents.

3- The design process and parametric studies are adequate.

4- The conclusion shows the obtained results and contribution.

5- References are up to date.

Author Response

Dear Reviewer-5,

Thank you very much for the useful comments on the paper. We have addressed these recommendations and uploaded a revised version of our paper. Please find below (Please see the attachment) our responses to the Reviewer-5 and a summary of our revision.

Reviewer-5 (Reviewer(s)' Comments to Author)

Comments to the Author

  1. The article is well written and organized.
  2. The abstract reflects the contents.
  3. The design process and parametric studies are adequate.
  4. The conclusion shows the obtained results and contribution.
  5. References are up to date

Dear Reviewer,

Thank you for your valuable comments and evaluations. We appreciate the valuable time you have devoted to review the paper. We have addressed the recommendations and uploaded a revised version of our paper. 

Reviewer 6 Report

1-      Since there are many papers in the literature on this subject, you should clearly state what is the difference between the proposed antenna structure and previously published methods/structures and why your method/structure is better/improved. You have to make it very clear what is new in this work or any novelty or some originality should be established right in the Abstract as well as in the Introduction.

2-      In the abstract, the state that the directivity is -3.18, please check the negative sign, it is incorrect.

3-      Some sentences are long and misleading, for example, the sentence in line 110 is not understandable. Please revise the technical writing throughout all manuscript.

4-      In Fig.3, you presented the antenna performance in two cases; skin and air. In the air case, please simulate using CST and HFSS and add them to the legend.

5-      The size of the SMA connector is comparable to the size of the antenna, I think the presence of the connector will affect the antenna's performance. It is better to use another type of small connector to avoid this effect.

6-      The radiation pattern experimental results are missing, it is a very important parameter for any antenna, please include it in the manuscript. The measurement setup should be included.

7-      You didn’t mention the gain value at your operating frequency, please discuss about simulation and experimental gain. Also, please justify if there is any discrepancy.

Author Response

Dear Reviewer-6,

Thank you very much for the useful comments on the paper. We have addressed these recommendations and uploaded a revised version of our paper. Please find below (Please see the attachment) our responses to the Reviewer-6 and a summary of our revision.

Reviewer-6 (Reviewer(s)' Comments to Author)

Comments to the Author

  1. Since there are many papers in the literature on this subject, you should clearly state what is the difference between the proposed antenna structure and previously published methods/structures and why your method/structure is better/improved. You have to make it very clear what is new in this work or any novelty or some originality should be established right in the Abstract as well as in the Introduction.

The novelty of this paper is that proposed antenna has a unique two-armed square spiral geometry along with a compact size of 10 x 10 x 2.5 mm2 (this is also seen in Table 3, which compares recent literature). Another novelty aspect of the study is that the performance of the proposed antenna design (S11, Radiation Pattern, SAR performance etc.)  in a realistic homogeneous 3D rat model is also considered in the scope of the study.

In line with your suggestion, the following paragraph has been added in the Abstract as well as in the Introduction.

“The novelty of this paper is that proposed antenna has unique two-armed square spiral geometry along with a compact size. Also, an important contribution of the paper is the consideration of the radiation performance of the proposed antenna design in a realistic homogeneous 3D rat model.”

  1. In the abstract, the state that the directivity is -3.18, please check the negative sign, it is incorrect.

Thank you for your careful review and suggestion. In the abstract, the negative sign in front of the directivity has been removed.

  1. Some sentences are long and misleading, for example, the sentence in line 110 is not understandable. Please revise the technical writing throughout all manuscript.

In line with your suggestion, the relevant sentence and the entire paper were reviewed and reconsidered.

  1. In Fig.3, you presented the antenna performance in two cases; skin and air. In the air case, please simulate using CST and HFSS and add them to the legend.

Based on your suggestion, HFSS results for air simulations are also included in
Figure 3.

  1. The size of the SMA connector is comparable to the size of the antenna, I think the presence of the connector will affect the antenna's performance. It is better to use another type of small connector to avoid this effect.

You are right in your comment. The connectors we use are high quality 50 ohm Huber+Suhner SMA type connectors. Until now, we have carried out similar implant antenna designs and fabrications, and we have not encountered any disruptive effects. We do not currently have small connectors of the type you mentioned. Since the order and shipping processes are quite long, it is not possible for us to try this at this time. We hope you will greet us with understanding.

  1. The radiation pattern experimental results are missing, it is a very important parameter for any antenna, please include it in the manuscript. The measurement setup should be included.

Unfortunately, we could not perform the radiation pattern measurements, since the non-isolated laboratory environment required for the measurement of the radiation pattern, especially for the measurement in liquid, is not available.

  1. You didn’t mention the gain value at your operating frequency, please discuss about simulation and experimental gain. Also, please justify if there is any discrepancy.

It is commonly known that dielectric loading effect, high attenuation due to the conductive nature of tissues, and reflections from the surface of the tissue drastically affect the antenna performance, resulting in low gains and low radiation efficiencies (< 1%). Besides, when the studies on implant antennas were examined, it was seen that the efficiency parameter of the antenna was not included. For this reason, the efficiency parameter is not included in the paper.

In addition, the following sentences are added to line 209:

“Also, the calculated efficiency and gain values are 0.13% and -18.41 dBi, respectively. It is known that the efficiency value is quite low (less than 1%) due to the high tissue loss and the miniature size of the antenna. Compared with similar studies in the literature [20-24], the efficiency of the proposed antenna is acceptable.”

Round 2

Reviewer 6 Report

I have no other comments, I recommend accepting this manuscript in this form.